# Monitoring of Chlorpyrifos Residues in Corn Oil Based on Raman Spectral Deep-Learning Model

**DOI:** 10.3390/foods12122402

**Published:** 2023-06-17

**Authors:** Yingchao Xue, Hui Jiang

**Affiliations:** School of Electrical and Information Engineering, Jiangsu University, Zhenjiang 212013, China; xueyingchao629@163.com

**Keywords:** Raman spectroscopy, corn oil, deep learning, chlorpyrifos, quantitative analysis

## Abstract

This study presents a novel method for the quantitative detection of residual chlorpyrifos in corn oil through Raman spectroscopy using a combined long short-term memory network (LSTM) and convolutional neural network (CNN) architecture. The QE Pro Raman+ spectrometer was employed to collect Raman spectra of corn oil samples with varying concentrations of chlorpyrifos residues. A deep-learning model based on LSTM combined with a CNN structure was designed to realize feature self-learning and model training of Raman spectra of corn oil samples. In the study, it was discovered that the LSTM-CNN model has superior generalization performance compared to both the LSTM and CNN models. The root-mean-square error of prediction (RMSEP) of the LSTM-CNN model is 12.3 mg·kg^−1^, the coefficient of determination ($R_{P}^{2}$) is 0.90, and the calculation of the relative prediction deviation (RPD) results in a value of 3.2. The study demonstrates that the deep-learning network based on an LSTM-CNN structure can achieve feature self-learning and multivariate model calibration for Raman spectra without preprocessing. The results of this study present an innovative approach for chemometric analysis using Raman spectroscopy.

## 1. Introduction

Chlorpyrifos has the chemical name O, O-diethyl-O-(3,5,6-trichloro-2-pyridyl) phosphorothioate, with a molecular formula of C_9_H_11_Cl_3_NO_3_PS and a molecular weight of 350.6. Chlorpyrifos is a white crystalline granular substance with a slight mercaptan odor [1]. It is a non-systemic broad-spectrum insecticide and acaricide, and is also an ideal insecticide for oil crops [2]. Chlorpyrifos is an acetylcholinesterase inhibitor and one of the organophosphorus insecticides. It has a controlling effect on pest problems of economic crops such as corn, soybeans, and peanuts. It can inhibit the activity of acetylcholinesterase in the nervous system of pests, thus disrupting their normal nerve impulse conduction [3]. Therefore, Chlorpyrifos has been widely used worldwide to increase the yield of various economic crops. However, due to its widespread use and persistence, it accumulates in the environment and may migrate to targeted plant bodies, affecting their physiological metabolism and crop quality [4]. This can easily lead to excessive residues of Chlorpyrifos in the subsequent processing of oil crops such as corn and soybeans, thereby reducing the quality of edible oil and posing risks to the physical and mental health of consumers.

Edible oil is a crucial consumer item that supplies the human body with vital nutrients such as energy and essential fatty acids. Moreover, it aids in the absorption of fat-soluble vitamins while also performing several essential functions in the body [5]. In 2020, the national consumption of vegetable oil exceeded 40 million tons for the first time. By 2021, the total consumption of vegetable oil in the country reached 42.545 million tons, an increase of 4.5% year-on-year, and the overall consumption scale reached a new high. During the planting process of oil crops, they are often attacked by pests and diseases, making pesticide control essential [6]. However, due to the high fat content in edible oil, fat-soluble pesticides can easily accumulate during the processing of edible oil and enter the human body through dietary consumption, posing a threat to consumers’ health. Therefore, it is necessary to conduct pesticide residue testing on edible oil. In recent years, the issue of pesticide residues in edible oil has received increasing attention from pesticide management departments, and chlorpyrifos, dimethoate, imidacloprid, and methomyl are common pesticide residues in corn oil [7]. The current methods for pesticide residue detection typically involve gas chromatography [8], gas chromatography–mass spectrometry [9], high-performance liquid chromatography [10], liquid chromatography–mass spectrometry [11], and rapid detection methods such as enzyme inhibition [12], immunoassay [13], and biosensors [14]. Among them, rapid detection methods can only be used as screening methods, and qualitative and quantitative results cannot be used as a basis for determination. Chromatography methods have the disadvantages of tedious qualitative procedures (usually requiring dual-column qualitative analysis), significant interference from sample matrix, and limitations in detection parameters. Although single-stage mass spectrometry is fast in qualitative analysis during detection, it has the drawbacks of high matrix interference, high detection limits, and inaccurate quantification [15]. Therefore, it is essential to develop a novel approach that can quickly and precisely detect pesticide residues in edible oils without causing pollution.

Raman spectroscopic analysis technology is a rapid and environmentally friendly, non-destructive method for detection, which has been applied to the qualitative and quantitative analysis of pesticide residues in agricultural products and foods [16]. Raman spectroscopy, also known as the Raman effect, was first discovered by Indian scientist C.V. Raman, and is a form of scattering spectroscopy [17]. By analyzing the spectrum of scattered light, structural information relating to the molecular vibration, rotation, and energy levels can be obtained, which enables the identification and characterization of substances. With the development of Raman signal enhancement techniques, the complexity of Raman spectroscopy has been greatly reduced, making it widely applicable in various fields such as food, agriculture, materials, and the environment. In the quality analysis of edible oil, Raman spectroscopy, combined with chemometrics methods, has been successfully applied [18,19,20]. These studies illustrate the potential of Raman spectroscopy for the qualitative analysis of food oil quality and adulteration. However, there is limited research on quantitatively analyzing pesticide residues, particularly chlorpyrifos residues in corn oil using Raman spectroscopy.

In addition, in Raman spectroscopy multivariate analysis, the detection of pesticide residues in agricultural products or food is generally studied using the full-band or wide-band method. Due to the large number of variables contained in Raman spectroscopy data, these variables not only reflect the chemical composition and content of the measured sample, but also the spectral response caused by factors such as the temperature, surface texture, color, and density of the sample. These noises will have a certain impact on the spectral analysis. Usually, the first step in processing different types of spectra is preprocessing. We usually choose multiple preprocessing methods based on our own experience to process and analyze the data. However, due to the influence of multiple factors, the generalization ability of this method is poor. It often has significant effects on the current dataset, but when the dataset is changed, the effect will sharply decline. In recent years, deep-learning algorithms with deep structures have shown great advantages in big data processing, especially in processing two-dimensional image data, and have made breakthrough progress. Raman spectroscopy data generally have the characteristics of wide spectral bandwidth, high dimensionality, severe spectral peak overlap, and complex internal information features. In processing complex information, deep-learning networks have better effects. In spectral chemometrics, the use of deep learning is currently experiencing significant advancements [21,22,23]. Based on the aforementioned analysis, this research aims to utilize samples of corn oil containing different concentrations of chlorpyrifos for the purpose of studying its effects. The objective is to design an effective deep-learning network architecture that enables the self-learning of features and model training using Raman spectra of corn oil samples. The ultimate goal is to accurately quantify the presence of chlorpyrifos residues in corn oil. This study’s development can offer a fast and precise alternative for the quality supervision department of grain and oil to monitor pesticide residues in edible oil.

## 2. Experimental Materials and Methods

### 2.1. Obtaining and Preparing Samples for Experimentation

Chlorpyrifos standard(concentration greater than 99%) and N-hexane (chromatographic grade) was obtained from Shanghai Aladdin Company, which is located in Shanghai, China. Nine kinds of corn oil samples were purchased from JD.com, and the brands were Happy swallow, Jiusan, Fortune, Golden dragon fish, Knife, Long Flower, Nissin, COFCO, and Lion&Globe.

When preparing edible oil samples, first, dissolve 250 mg of chlorpyrifos standard in n-hexane solvent. Next, prepare a series of 21 chlorpyrifos standard solutions with varying concentrations, specifically 1800, 500, 400, 300, 180, 150, 100, 50, 40, 30, 15, 12, 10, 5, 4, 3, 1.8, 1, 0.8, 0.5, and 0.3 mg/kg. Lastly, dissolve the chlorpyrifos standard solutions of different concentrations in a 1:9 mass ratio into the corn oil purchased from the supermarket, resulting in 189 corn oil samples with diverse chlorpyrifos levels. The concentration gradients of samples are 180, 50, 40, 30, 18, 15, 10, 5, 4, 3, 1.5, 1.2, 1, 0.5, 0.4, 0.3, 0.18, 0.1, 0.08, 0.05, and 0.03 mg/kg.

### 2.2. Experimental Apparatus

Raman spectra of the corn oil samples were collected using an Ocean Optics QE Pro Raman spectrometer obtained from Ocean Optics Inc., located in Dunedin, FL, USA. Raman + laser Raman spectrometer and spectral data were recorded with the OceanView software (v2.0.8). The quartz cuvette used for the samples had a 5 mm width.

### 2.3. Spectral Sampling of Samples

The instrument was set for spectral acquisition with a 532 nm laser source at 300 mW power and an integration time of 1000 ms. The spectral scan was performed in the range of 84–4540 cm^−1^ at an ambient temperature of approximately 25 °C. To acquire the raw Raman spectra of the samples, three measurements were taken for each sample and an average spectrum was generated from those measurements. Figure 1 shows the raw Raman spectra of the samples.

### 2.4. Data Analysis Methods

#### 2.4.1. Overview of Convolutional Neural Networks (CNN)

CNN has been widely used in the field of spectral chemometrics analysis recently [24]. Convolutional layers, activation layers, pooling layers, and fully connected layers are all essential components of a CNN. The convolutional layer performs cross-correlation operations between the input and convolutional kernel weights, and produces an output after adding scalar biases. Therefore, the two trained parameters in the convolutional layer are the convolutional kernel weights and scalar biases. Just as we randomly initialize fully connected layers before training models based on convolutional layers, we also randomly initialize convolutional kernel weights when training models based on convolutional layers. The activation layer is used to perform nonlinear transformations on the output signals of the convolutional layer and the fully connected layer. Generally, an activation function is added after the convolutional layer and the fully connected layer to improve the model’s nonlinear expressive ability. Studies have shown that different activation functions have a significant impact on the performance of the model, and the sigmoid, hyperbolic tangent (tanh), and rectified linear unit (ReLU) activation functions are currently the most common. Pooling, also known as subsampling or downsampling, is mainly used for feature dimensionality reduction, data compression, and reducing the number of parameters, while improving the model’s fault tolerance and reducing overfitting. After several convolutional and activation and pooling operations, we finally reach the output layer, where the model connects the learned high-quality features. Introducing dropout operations before the fully connected layer can prevent overfitting that may occur if the number of neurons is too large and the learning capability is strong. This method randomly deletes some neurons in the neural network, enhancing its overall performance. Local standardization, data augmentation, and other operations can also be performed to increase robustness. To determine the appropriate activation function and neuron count, we consider the specific prediction task at hand. The requirements for different predictions will vary, and, as a result, the activation function and neuron count will also differ.

Based on the above analysis, the structure of the CNN network for analyzing chlorpyrifos residues in corn oil is displayed in Figure 2. A 1D-CNN model was employed to create a detection system for forecasting the presence of chlorpyrifos in corn oil samples within this architecture. To showcase the benefits of deep-learning algorithms in spectral analysis, Raman spectroscopic data underwent preprocessing via normalization. The study utilized a CNN architecture that included three convolutional layers and three pooling layers to extract spectral data features. A fully connected layer and an output layer were then connected to establish the CNN model with the activation function set to Relu. The purpose of this is to enable the spectral feature points (a total of 996 feature points) to fully reflect the residue of chlorpyrifos. The model parameters were determined through repeated debugging, taking into account the situations of overfitting and underfitting. In Figure 2, blue represents the input, green represents the output after convolution, pink represents the output after pooling, and white represents the convolution or maximum pooling operation. Orange yellow represents flatten and full connection.

#### 2.4.2. Overview of Long Short-Term Memory Neural Networks

The long short-term memory network (LSTM) is an improvement over the recurrent neural network (RNN), which can only remember short-term sequential features [25]. It is a type of model that excels at learning time-series features in deep-learning technology. Since RNN is good at short-term memory, the model’s performance will be poorer for longer data sequences, and network depth will lead to problems such as gradient disappearance that can prevent successful model training. Therefore, this study chose the LSTM model as an improvement method for RNN to perform feature self-learning and model calibration on Raman spectra. The LSTM network structure designed for quantitative analysis of the residual amount of chlorpyrifos in corn oil is shown in Figure 3.

Figure 3 illustrates the internal structure of the LSTM unit within the network in detail. Here, t represents the time step; $x_{t}$ and $h_{t}$ are the input and output vectors of the hidden layer, respectively; and $c_{t}$ is the memory cell. Each LSTM structure unit mainly includes three types of gate control: input gate, forget gate, and output gate. The internal working mechanism of the LSTM memory cell can be summarized by the following equations:(1)$$f_{t}=\sigma (W_{f}\times [x_{t},h_{t-1}]+b_{f})$$
(2)$$i_{t}=\sigma (W_{i}\times [x_{t},h_{t-1}]+b_{i})$$
(3)$$O_{t}=\sigma (W_{o}\times [x_{t},h_{t-1}]+b_{o})$$
(4)$$\tilde{C_{t}}=tanh(W_{c}\times [x_{t},h_{t-1}]+b_{c})$$
(5)$$C_{t}=(f_{t}\times C_{t-1}+i_{t}\times \tilde{C_{t}})$$
(6)$$h_{t}=tanh(C_{t})\times O_{t}$$
where σ(x) is the sigmoid function mentioned above, and W and b denote the weight matrix and bias vector of the network. The implementation of this generic expression usually relies on the sigmoid function and the dot product operation. Gating is similar to the role of the fully connected layer in that it helps LSTM units to store and update information more efficiently. More specifically, input gating serves to control the occupancy problem, that is, how much of the input data at the current moment of the network can be stored in the memory cell; forget gating is used to decide the information trade-off problem, that is, which information is desired to be remembered or forgotten; and output gating is used to control the specific output at the current moment by adjusting the memory cell c. In this model, detection models are constructed using LSTM networks to predict the chlorpyrifos content of the samples. Similar to the CNN model, the only preprocessing performed on the Raman spectral data is normalization. To extract spectral sequence information, the study employed two LSTM layers in the network. An output layer was then connected to generate the results. The number of hidden units included in the LSTM prediction network model is 80, and the maximum number of training cycles is set to 1600.

#### 2.4.3. Overview of LSTM-CNN Neural Networks

The prediction results obtained by CNN and LSTM models for the same dataset are different. This is because different prediction methods have different ideas and different angles for dealing with the same problem. Each model has its own advantages and disadvantages, so, in usual data analysis, a variety of different algorithms will be tried, and then weighed according to the subsequent verification effect, project resources, and value, and the final choice will be made.

The aim of constructing the model is to enhance the accuracy of predictions and minimize the discrepancy between predicted values and actual values. Therefore, using the raw data as input can only predict the LSTM neural network based on the raw features of the input data, which is not accurate enough. As LSTM solves the problem of long-term dependencies and can be used for time-series prediction, and CNN is better at feature extraction, the data processed by LSTM can be used as the input of CNN. Figure 4 depicts the architecture of an LSTM-CNN model that can be designed for the same purpose. The overall architecture of the model consists of four parts, including LSTM layer, convolutional layer, pooling layer, and fully connected layer. The LSTM and CNN modules are designed as before, but the LSTM is mainly responsible for Raman spectroscopy feature learning, and the CNN module is mainly responsible for multivariate model calibration. In this way, the LSTM-CNN structure can fully leverage the strengths of LSTM and CNN, thus achieving the goal of improving prediction accuracy.

#### 2.4.4. Diagnosis of Nonlinearity

In this study, the relationship between Raman spectroscopy signals and concentrations of the pesticide chlorpyrifos is complex. To assess whether a linear relationship exists between the Raman spectroscopy signals and the target concentration, this study assumed linearity and used a partial least squares (PLS) model to predict the chlorpyrifos content in the samples. The mean partial residual plot (APaRP) method recommended by Mallows was then used to diagnose nonlinearity [26]. We used quantitative numerical tools to determine the nonlinearity based on the APaRP method, with parameters including $n_{+}$ (the total number of positive consecutive residual sequences), $n_{-}$ (the total number of negative consecutive residual sequences), and *u* (the number of runs). In general, the following approximations yield satisfactory results when $n_{+}$ > 10 and $n_{-}$ > 10.
(7)$$\mu =\frac{2n_{+}n_{-}}{n_{+}+n_{-}}+1$$
(8)$$\sigma^{2}=\frac{2n_{+}n_{-}\left(2n_{+}n_{-}-n_{+}-n_{-}\right)}{{\left(n_{+}+n_{-}\right)}^{2}\left(n_{+}+n_{-}-1\right)}$$
(9)$$z=\frac{u-\mu +0.5}{\sigma }$$
where *z* represents the desired randomness measure, and, when |*z*| exceeds 1.96, it indicates a nonlinear relationship between the two datasets.

### 2.5. Model Evaluation

The evaluation of Raman spectral model usually uses root-mean-square error (RMSE), coefficient of termination (R^2^), relative percent deviation (RPD) and other indicators for comprehensive evaluation. These indices can evaluate the reliability, prediction ability, and fitting of the model performance, and are of great significance for the establishment of accurate and reliable Raman spectral models. The formulae for RMSEP and $R_{P}^{2}$ are similar to RMSEC and $R_{C}^{2}$. Here are the equations for calculating these measures:(10)$$\mathrm{RMSEC}=\sqrt{\frac{\sum_{i=1}^{n_{c}}{\left(y_{i}-\hat{y_{l}}\right)}^{2}}{n_{c}}}$$
(11)$$R_{C}^{2}=1-\frac{\sum_{i=1}^{n_{c}}{\left(y_{i}-\hat{y_{l}}\right)}^{2}}{\sum_{i=1}^{n_{c}}{\left(y_{i}-\overset{-}{y_{l}}\right)}^{2}}$$
(12)$$\mathrm{RPD}=\frac{SD}{RMSEP}$$

The formula involves several variables, such as the measured value ($y_{i}$), predicted value ($\hat{y_{i}}$), and average value ($\overset{-}{y_{i}}$) of the calibration set. Moreover, $n_{c}$ denotes the calibration set’s size, and SD represents the prediction set’s standard deviation.

### 2.6. Software

The study employed PyTorch 1.5.0 (Python 3.6.9) to implement all deep-learning algorithms. The experimental data were processed on a system with an Intel i7 12700 CPU, 16 GB of memory, an RTX3080 GPU with 10 GB of memory, and the Windows 10 operating system.

## 3. Results

### 3.1. Subsection

For data analysis, seven samples from the training set’s identical concentration gradient were randomly chosen, while the remaining two samples were designated as the prediction set. Therefore, the training set and prediction set contained 147 and 42 samples, respectively. The statistical outcomes of the corn oil samples’ chlorpyrifos content within the training set and prediction set are presented in Table 1. Table 1 analysis indicates that the partitioning of the dataset is reasonable and suitable for model training.

### 3.2. Results of Diagnosis of Nonlinearity

Table 2 presents the test results, with a |*z*| value of 24.1 indicating a nonlinear relationship between Raman spectroscopy signals and chlorpyrifos residue values. In order to accurately predict the residual amount of chlorpyrifos in corn oil, this study employed nonlinear algorithms 1D-CNN, LSTM, and LSTM-CNN to establish a regression model for predicting chlorpyrifos residue in corn oil.

The molecular structure of chlorpyrifos is complex, containing multiple functional groups such as phosphorus, oxygen, and sulfur. The presence of these functional groups leads to complex vibrational modes, resulting in complex Raman spectra. The nonlinearity between the Raman spectral signal and the concentration of chlorpyrifos can be attributed to the addition effect. The addition effect arises from the fact that the Raman signals of different functional groups in the chlorpyrifos residue are additive, meaning that the total Raman signal of the residue is the sum of the Raman signals of each functional group. The interaction between different functional groups may lead to changes in the Raman signal, making this effect more complex. Additionally, when preparing the chlorpyrifos solution, n-hexane was added to promote better mixing with the corn oil. Other compounds present in the sample may interfere with the Raman spectra, leading to nonlinearity.

### 3.3. Results and Comparison of Different Models

The training outcomes of various deep-learning models with distinct network architectures are depicted in Figure 5. As depicted in Figure 5, the loss function for each model decreases as the number of training iterations increases. It can be inferred that all models aim to identify the characteristics that correspond to the residual level of chlorpyrifos in corn oil. Figure 5a–c were further examined, and it was discovered that each model’s training process gradually stabilizes, with the models reaching a stable point at approximately the 1400th iteration. The CNN and LSTM-CNN models have been continuously reducing their loss values, with a fast convergence speed that quickly reaches a stable state. Notably, the loss value of the LSTM-CNN model is marginally lower than that of the CNN model, indicating that the LSTM-CNN model has a better performance in the detection of chlorpyrifos residual content in corn oil. The loss value of the LSTM model oscillates back and forth continuously and gradually reaches a stable state after 300 epochs. This implies that the LSTM model’s complexity is excessive, causing the loss value to oscillate around the minimum value during the early stages of training but failing to attain it. With an increase in the number of training iterations, the LSTM model stabilizes gradually.

The results in Table 3 demonstrate that the $R_{P}^{2}$ values for both sample sets are greater than 0.87 for all models, including the CNN, LSTM, and LSTM-CNN models. Hence, the deep-learning network models employed in this study are appropriate for the multivariate analysis of Raman spectra in corn oil samples and satisfy the prerequisites of spectroscopic chemometrics analysis. Overall, the LSTM-CNN model exhibits superior performance in both sample sets compared to the CNN and LSTM models, with an $R_{P}^{2}$ value of 0.90 in the prediction set. This value is slightly higher than that of the CNN model by 0.029 and slightly higher than that of the LSTM model by 0.025; the RMSEP and RPD values of the proposed model outperform both the CNN and LSTM models, with the former being nearly 1.7 and 1.5 lower, respectively. Additionally, the RPD value is nearly 0.38 and 0.34 higher than that of the CNN and LSTM models, respectively. Therefore, we can conclude that the LSTM-CNN model performs better than the CNN and LSTM models in processing Raman spectroscopy sequence data.

## 4. Conclusions

This study validated the potential application of the LSTM-CNN deep-learning model in the chemometric analysis of Raman spectra. In the multivariate analysis of the Raman spectra of corn oil samples, the LSTM-CNN model exhibited superior generalization performance compared to the LSTM and CNN models, with the $R_{P}^{2}$ exceeding 0.9 and the RPD of 3.2. The results demonstrate the potential of this method for the rapid and accurate detection of pesticide residues in edible oils. The findings of this study present a novel reference method for the chemometric analysis of Raman spectra, with the potential for practical applications in various industries, such as food safety, pharmaceuticals, and environmental monitoring. The LSTM-CNN model’s enhanced performance and efficiency could lead to the more accurate and rapid detection of adulterations or contaminants, ultimately benefiting public health and safety.

## Figures and Tables

**Figure 1 foods-12-02402-f001:**
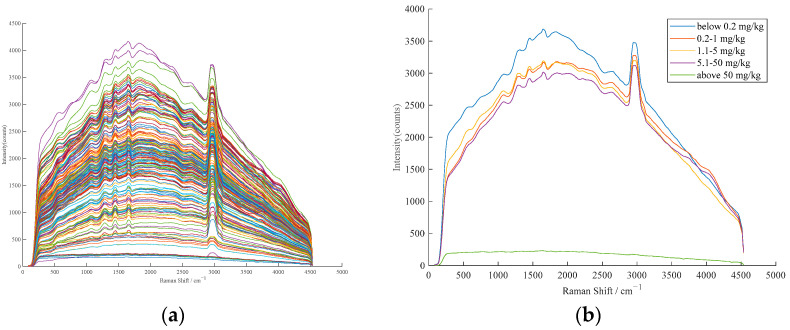
(**a**) Raw Raman spectra of 189 corn oil samples; and (**b**) raw Raman spectra of corn oil with different chlorpyrifos content.

**Figure 2 foods-12-02402-f002:**
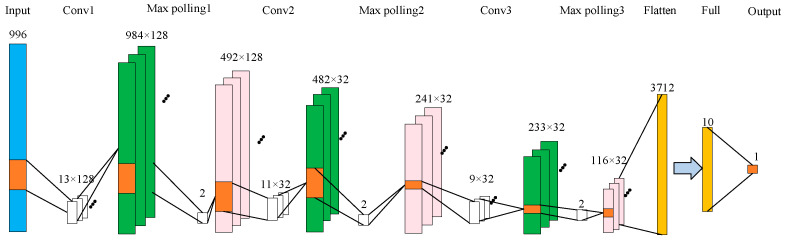
Network structure of CNN.

**Figure 3 foods-12-02402-f003:**
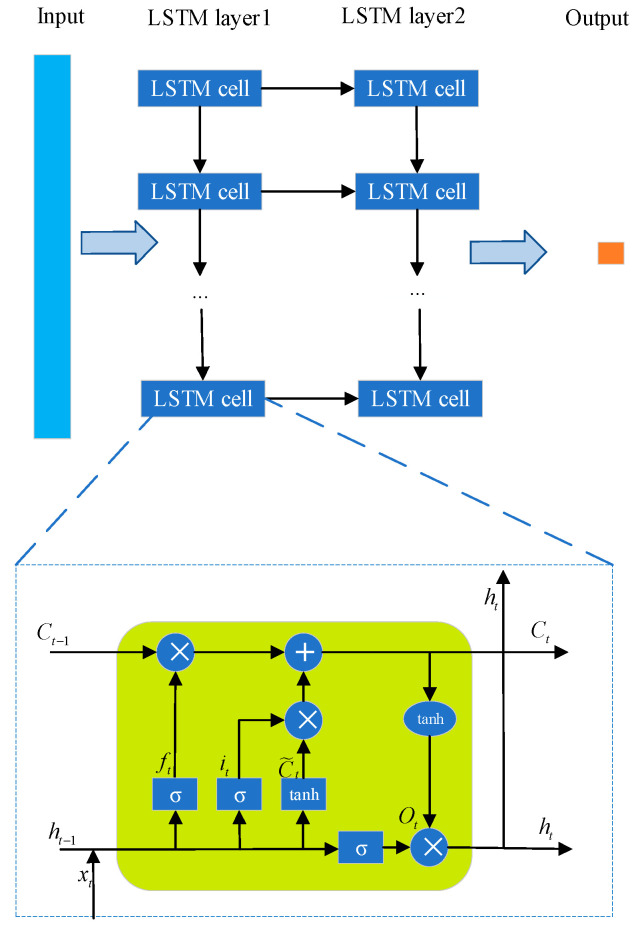
Network structure of LSTM.

**Figure 4 foods-12-02402-f004:**
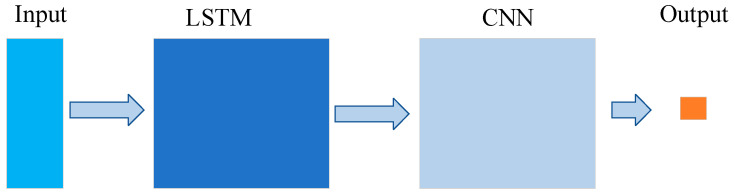
Network structure of LSTM-CNN.

**Figure 5 foods-12-02402-f005:**
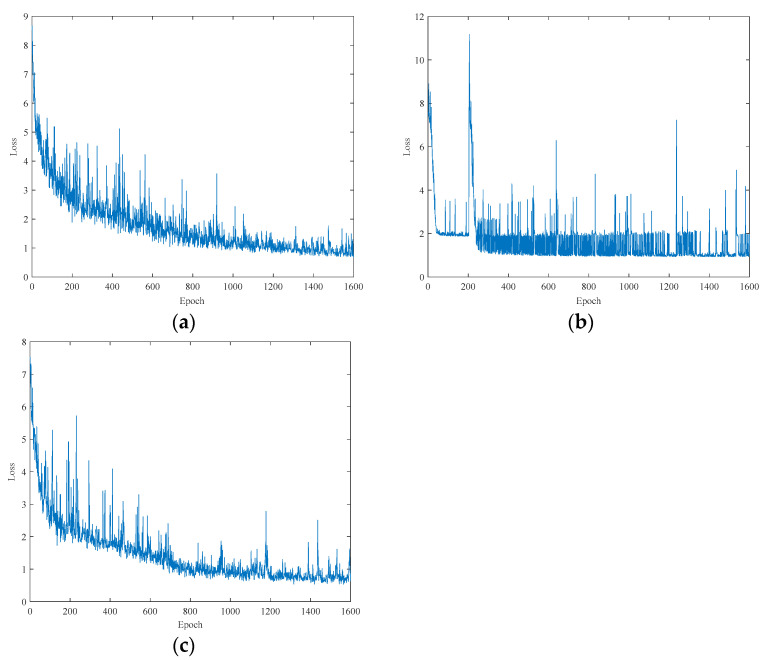
The loss of the (**a**) CNN, (**b**) LSTM, and (**c**) LSTM-CNN models.

**Table 1 foods-12-02402-t001:** Training set and prediction set reference measurement results of chlorpyrifos content in corn oil.

Subsets	Sample Number	Maximum/mg·kg^−1^	Minimum/mg·kg^−1^	Mean/mg·kg^−1^	Standard Deviation/mg·kg^−1^
Training set	147	180	0.03	17.2	39.1
Prediction set	42	180	0.03	17.2	39.5

**Table 2 foods-12-02402-t002:** The results of the runs test used to detect the non-linearity of Raman spectral signals and chlorpyrifos residue values by the APaRPs method.

$n_{+}$	$n_{-}$	u	μ	σ	$\left|z\right|$	Conclusion
223.9	359.4	1	276.6	11.4	24.1	Nonlinearity

$n_{+}$: the total number of positive consecutive residual sequences. $n_{-}$: the total number of negative consecutive residual sequences. u: the number of runs.

**Table 3 foods-12-02402-t003:** Evaluation of three neural networks.

Model	RMSEC/mg kg^−1^	$R_{C}^{2}$	RMSEP/mg kg^−1^	$R_{P}^{2}$	RPD
CNN	11.5	0.91	14.0	0.87	2.8
LSTM	13.8	0.87	13.7	0.88	2.9
LSTM-CNN	11.5	0.91	12.3	0.90	3.2

## Data Availability

The data is currently classified and will be available in 2024 with permission from the project.

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
