# Peer review of "Monitoring of Chlorpyrifos Residues in Corn Oil Based on Raman Spectral Deep-Learning Model"

_foods, 2023, doi:10.3390/foods12122402_

Round 1

Reviewer 1 Report

The manuscript titled “Monitoring of chlorpyrifos residues in corn oil based on Raman spectral deep learning model” by Xue and Jiang with ultimate goal is to accurately quantify the presence of chlorpyrifos residues in corn oil through designing an effective deep learning network architecture that enables self-learning of features and model training using Raman spectra of corn oil samples.

Some comments are raised in the attached manuscript.

Also, why there is no data regarding the residue values of chlorpyrifos from concentrations spiked with corn oil using your method? as the manuscript title is Monitoring of chlorpyrifos residues in corn oil based on ….

To make a judgment on the new method, it should be compared with a registered method. In this case LC-/GC-MS might be suitable.

Minor editing of English language required

Author Response

Reviewer1 #

1: In Line 25, "drug" this item not suitable for chlorpyrifos.

Response: Thank you for bringing this to our attention. We apologize for any confusion that may have been caused by the use of the term "drug" in the original text. As you correctly pointed out, chlorpyrifos is not a drug, but rather a pesticide and insecticide. Following your suggestion, we have revised it in the revised manuscript, please see the Line 25 at Page 1.

2.: In Section 2.2, please mention the origin of this instrument and address.

Response: Thank you for your feedback. We apologize for not including the origin and address of the instrument used in the experiment. The Ocean Optics QE Pro Raman+ laser Raman spectrometer was obtained from Ocean Optics Inc., located in Dunedin, Florida, USA. Please see the Section 2.2 in the revised manuscript.

3: In Figure 1, is this for one sample or many samples? the x axis title is not right. please remove.

Response: Thank you for your valuable feedback on Figure 1. We apologize for any confusion caused by the x-axis title. Upon careful consideration, we agree with your observation that the x-axis title may not accurately represent the data. Therefore, we have decided to modify the x-axis title to avoid any potential misinterpretation. Please see the Line 128 at Page3 in the revised manuscript.

Regarding your question about whether the figure represents one sample or many samples, we appreciate your inquiry. Figure 1 represents the results obtained from multiple samples collected in our study. We have clarified this information in the revised caption of the figure to provide better context for the readers. Please see the Lines 128-129 at Page3 in the revised manuscript.

Once again, we appreciate your insightful feedback, which has helped us improve the clarity and accuracy of our work. We have made the necessary revisions based on your suggestions and believe that the updated Figure 1 and its caption will enhance the overall quality of our paper. Thank you for your time and valuable input.

4: In line 279, the results in Table 2 are ambiguous. Please make an explanation of the parameters in the table footnote.

Response: Thank you for your feedback regarding Table 2. We have provided a detailed explanation of the parameters in the Table 2, Lines 279-281 in the revised manuscript. If you have any further questions or concerns, please do not hesitate to let us know. We appreciate your input and strive to improve the quality of our research.

Reviewer 2 Report

The manuscript by Xue and Jiang introduces a method for quantitative detection of residual chlorpyrifos in corn oil using Raman spectroscopy. The researchers employed a combination of long short-term memory network (LSTM) and convolutional neural network (CNN) architecture. Raman spectra of corn oil samples with varying chlorpyrifos concentrations were collected using the QE Pro Raman+ spectrometer. The LSTM-CNN model demonstrated superior generalization performance compared to the individual LSTM and CNN models. The root mean square error of prediction (RMSEP) for the LSTM-CNN model was 12.3 mg·kg-1, the coefficient of determination was 0.90, and the relative prediction deviation was 3.2. The study presented in manuscript highlights that the LSTM-CNN network can autonomously learn features and calibrate multivariate models for Raman spectra.

- why Authors used Raman spectrocopy and did not tested SERS (Surface Enhanced Raman Spectroscopy)? It could give better results.

- In introduction Authors wrote: "However, there is limited research on quantitatively analyzing pesticide residues, particularly chlorpyrifos residues in corn oil using Raman spectroscopy. " Does in mean that Authors demonstrate this approach for testing chlorpyrifon for the first time? was is testes by Raman before? or using ML?

-more information is needed on spectral sampling of samples. Did Authors did any postprocessing to the spectra except averaging? Figure 1 should give some information about the samples, maybe Authors could present additional Figure where they compare single spectra of two or three oils.

- in Conclusions please elaborate more on the results. Are they better than other methods? faster? is there any potential for practical application?

- line 125, cm-1, please check Celsius degree. Please re-read whole manuscript.

Some mino mistakes, Authors sholuld re-read it before submitting again.

Author Response

1: In introduction Authors wrote: "However, there is limited research on quantitatively analyzing pesticide residues, particularly chlorpyrifos residues in corn oil using Raman spectroscopy. " Does in mean that Authors demonstrate this approach for testing chlorpyrifos for the first time? was is testes by Raman before? or using ML?

Response: Yes, to our knowledge, this study is the first to quantitatively analyze chlorpyrifos residues in corn oil using Raman spectroscopy and deep learning algorithms. Previous studies on the detection of pesticide residues in edible oils using Raman spectroscopy were mainly qualitative analyses or did not involve chlorpyrifos. Some studies have applied chemometrics methods to Raman spectra for the quantitative analysis of pesticides in agricultural products and foods. However, research on the quantitative analysis of pesticide residues, especially chlorpyrifos pesticides, in edible oils using Raman spectroscopy is still lacking. Therefore, this study aims to explore the potential of Raman spectroscopy for the quantitative analysis of chlorpyrifos residues in corn oil.

2: More information is needed on spectral sampling of samples. Did Authors did any postprocessing to the spectra except averaging? Figure 1 should give some information about the samples, maybe Authors could present additional Figure where they compare single spectra of two or three oils.

Response: Dear Reviewer, thank you for your insightful comments and suggestions. We appreciate your interest in our work and will address your concerns as follows. First, no other preprocessing was performed on the Raman spectra except for averaging. Because the use of deep learning algorithms can obtain more features of the original image. Second, we agree with your suggestion to provide more information about the samples in Figure 1. We have created a new figure (Figure 1(b)) that presents this comparison.

We hope these revisions will address your concerns and improve the clarity of our manuscript. Thank you once again for your valuable feedback.

3: In Conclusions please elaborate more on the results. Are they better than other methods? faster? is there any potential for practical application?

Response: Thank you for your valuable feedback on our paper. We have carefully considered your suggestion and have made the following revisions to the Conclusions section, please see the Lines 326-333 at Page9 in the revised manuscript.

4: Line 125, cm-1, please check Celsius degree. Please re-read whole manuscript.

Response: Thank you for your feedback regarding our manuscript. We apologize for the error in our manuscript. We have fixed it on line 125 at Page3 in the revised manuscript.

We appreciate your careful review of our manuscript and will make sure to thoroughly re-read the entire manuscript to ensure that all errors are corrected and the manuscript meets the highest standards of scientific quality. If you have any further comments or suggestions, please do not hesitate to let us know. Thank you again for your valuable feedback.

Round 2

Reviewer 1 Report

No more comments.

Minor editing of English language required.

Reviewer 2 Report

The current manuscript can be accepted for publication.